# Sea Ice Image Classification Based on Heterogeneous Data Fusion and Deep Learning

**Yanling Han, Yekun Liu, Zhonghua Hong \*** **, Yun Zhang, Shuhu Yang and Jing Wang**

Key Laboratory of Fisheries Information, Ministry of Agriculture; Shanghai Marine Intelligent Information and Navigation Remote Sensing Engineering Technology Research Center, College of Information, Shanghai Ocean University, Shanghai 201306, China; ylhan@shou.edu.cn (Y.H.); m180701061@st.shou.edu.cn (Y.L.); y-zhang@shou.edu.cn (Y.Z.); shyang@shou.edu.cn (S.Y.); wangjing@shou.edu.cn (J.W.)

\* Correspondence: zhhong@shou.edu.cn

**Abstract:** Sea ice is one of the typical causes of marine disasters. Sea ice image classification is an important component of sea ice detection. Optical data contain rich spectral information, but they do not allow one to easily distinguish between ground objects with a similar spectrum and foreign objects with the same spectrum. Synthetic aperture radar (SAR) data contain rich texture information, but the data usually have a single source. The limitation of single-source data is that they do not allow for further improvements of the accuracy of remote sensing sea ice classification. In this paper, we propose a method for sea ice image classification based on deep learning and heterogeneous data fusion. Utilizing the advantages of convolutional neural networks (CNNs) in terms of depth feature extraction, we designed a deep learning network structure for SAR and optical images and achieve sea ice image classification through feature extraction and a feature-level fusion of heterogeneous data. For the SAR images, the improved spatial pyramid pooling (SPP) network was used and texture information on sea ice at different scales was extracted by depth. For the optical data, multi-level feature information on sea ice such as spatial and spectral information on different types of sea ice was extracted through a path aggregation network (PANet), which enabled low-level features to be fully utilized due to the gradual feature extraction of the convolution neural network. In order to verify the effectiveness of the method, two sets of heterogeneous sentinel satellite data were used for sea ice classification in the Hudson Bay area. The experimental results show that compared with the typical image classification methods and other heterogeneous data fusion methods, the method proposed in this paper fully integrates multi-scale and multi-level texture and spectral information from heterogeneous data and achieves a better classification effect (96.61%, 95.69%).

**Keywords:** sea ice; heterogeneous data; data fusion; feature information

## 1. Introduction

Sea ice, which accounts for 5–8% of the global ocean area, is the most prominent cause of marine disaster in polar seas and some high-dimensional regions. Polar sea ice anomalies affect atmospheric circulation, destroy the balance of fresh water, and affect the survival of organisms. Mid–high latitude sea ice disasters affect human marine fisheries, coastal construction, and manufacturing industries, and they also cause serious economic losses [1]. Therefore, sea ice detection has important research significance, and sea ice image classification is an important part of it.

It is necessary to obtain effective data in a timely manner for sea ice detection. Remote sensing technology provides an important means for large-scale sea ice detection. Traditional remote sensing detection data include SAR and optical remote sensing data with a high spatial resolution and high spectral resolution (such as MODIS, Sentinel-2, and Landsat). As an active microwave imaging radar, SAR has the characteristics of having an all-day, all-weather, and multi-perspective collection method with a strong penetration,

and its images contain rich texture information [2], achieving good results in sea ice classification [3–5]. With the continuous development of optical remote sensing technology, the multi/hyperspectral resolution of optical remote sensing images can now provide more detailed information in the spectral dimension, which provides important data support for the classification of sea ice images. At present, more and more optical images have been used in sea ice classification, such as MODIS optical data [6] and Landsat optical data [7], and good classification results have been achieved.

In recent years, deep learning has rapidly developed in computer vision. With artificial intelligence, the explosive development of all kinds of deep learning algorithms has also gradually begun to mature, and these algorithms have come to replace traditional classification algorithms, such as the support vector machine algorithm (support vector machine, SVM) [8], which has the problem of giving priority to shallow characteristics in feature extraction and thus neglecting the deeper characteristics. Deep learning models do not rely on manual design features and can extract features at different levels, including shallow, deep, and complex features, which allows for considerable achievements in computer vision image classification [9]. The convolutional neural network is an important deep learning algorithm. The AlexNet deep convolutional neural network proposed by Krizhevsky et al. [9] won the first prize in the image classification Contest of ILSVRC 2012 (ImageNet Large Scale Visual Recognition Challenge), and the error rate was reduced by about 10% compared to the traditional classification algorithm. Kaiming He et al. [10] proposed a deep residual network (Resnet) that can increase the number of layers in a network to hundreds in order to extract more information on image characteristics. In 2014, Kaiming He et al. proposed an SPP method [11] that can integrate features of different sizes. In 2018, Shu Liu et al. proposed the PANet model [12], emphasizing that information propagation between layers is very important in deep learning networks, and a similar Feature Pyramid Network (FPN) [13] has achieved excellent experimental results. The achievement of the convolutional neural network in image classification provides a new technical means for remote sensing image classification. In the field of remote sensing image classification, the convolutional neural network can be used to directly extract features of SAR images and optical images [14,15]. At the same time, the deep learning method also made a new breakthrough in the classification of sea ice. The deep learning model is used to classify sea ice, and the effect is significantly better compared to that of the traditional classification methods [16,17]. However, at present, most sea ice detection methods use only a single data source. Because single-source methods have limited information on the characteristics of images and another restriction pertaining to the imaging index, the expression of feature information is not comprehensive [18]. The inclusion of different types of ice in the detection of class differences between smaller and bigger ice increases the difficulty of the classification of sea ice. At the same time, the broad classification of sea ice as a "foreign body with different spectra" causes the ice model to be easily confused. The method also needs support for more types of feature information. Using a single data source therefore makes it difficult to further improve the detection accuracy of sea ice.

Based on the above research, SAR data and optical data are acquired by different sensors, and the information contained in the images is also different. SAR data are mostly single-band data, so it is difficult to distinguish between types of sea ice, but SAR images contain rich texture information. Optical data have more bands and contain rich spectral information, which can provide detailed data support for sea ice classification. However, there are certain limitations in terms of discriminating between classes with spectral similarity (such as gray and white ice). Therefore, combining the rich spectral features provided by optical remote sensing data and the advantages of SAR images in terms of texture features, this paper puts forward a method based on deep learning and the fusion of heterogeneous data from different sea ice image classification methods. We utilize the advantage of convolution neural networks in terms of the depth of the feature extraction that are designed for the depth of SAR images and the optical image network structure, learn through different source data feature extractions, and feature level fusion

sea ice image classification. For the SAR images, the improved SPP network is used to realize feature extraction at different scales in order to extract texture information of sea ice in depth. For the optical data, through the extraction of different types of PANet multi-level characteristics of spatial and spectral information on sea ice, convolutional neural network feature extraction is constructed step by step, thus adequately utilizing low-level features. Finally, features are extracted through the fusion of two models, making full use of the heterogeneous multi-scale data and multi-level classification of the depth characteristics of sea ice.

The rest of this paper is arranged as follows: The second section describes the design framework and algorithm of the proposed method; the third section introduces the experimental data and settings in detail, and the model parameters and experimental results are discussed and analyzed; and the fourth section summarizes the work presented in this paper.

## 2. Sea Ice Classification Method Based on Heterogeneous Data Fusion

### 2.1. Sea Ice Classification Framework Based on Heterogeneous Data Fusion

The framework of sea ice classification based on the heterogeneous data fusion proposed in this paper is shown in Figure 1, which mainly includes four parts, namely, SAR image feature extraction, optical image feature extraction, feature fusion, and sea ice classification and accuracy assessment. Firstly, SAR images and optical images are preprocessed. The SAR images are processed for thermal sound removal, spot removal, and geometric correction, whereas the optical images are mainly processed for atmospheric correction, radiometric calibration, and principal component analysis. The pre-processed image is resampled to a resolution of 10 m. Then the resampled SAR image is extracted with multi-scale features through the improved spatial pyramid network and the optical image is extracted with spatial and spectral information through the PANet network. Next, the two branches are fused to extract feature information, and the full connection layer is inputted through the Softmax classifier for classification. Finally, the confusion matrix is calculated through the overall accuracy and Kappa coefficient, and a classification accuracy assessment is conducted. At the same time, the suggested method is analyzed and compared to SVM, the two-branch CNN [19], the deep fusion model [20], and other methods.

### 2.2. Heterogeneous Data Fusion Network Model

The structure of the heterogeneous network model for data fusion is shown in Figure 2. The model includes a two-branch module and a module for SAR images, because sea ice in SAR images contains more abundant texture features. This article is based on the SPP model and puts forward the improved spatial pyramid pooling (ISPP) model in order to fully extract different scales of sea ice feature information Another module for optical images, the PANet network model, enhanced by a bottom-up path, makes full use of the low-level features and enhances the whole level between the low-level features and high-level features. It also shortens the path of information and further integrates the feature extraction of useful information at every level in order to enhance the characteristics of sea ice. The enhanced features extracted from the two branches were fused and inputted into the full connection layer. Finally, the results of sea ice classification were obtained by inputting them into the classifier.

#### 2.2.1. Improved SPP Model

The function of the SPP model is that it fuses different features at multiple scales, which can transform the feature graph of any size into a fixed-size feature vector that is inputted into the full connection layer. As shown in Figure 3 [11], the feature graph output of the convolution layer is then inputted into the SPP module, and a total of $(16 + 4 + 1) \times 256$ features can be outputted, where, $16 + 4 + 1$ represents the number of spatial bins, and 256 represents the number of convolution cores. In this way, multi-scale feature vectors are obtained.

## Image preprocessing

**SAR**

Thermal Noise Removal → Calibrate to sigma nought → Speckle filtering → Geometric Correction → S1 and S2 resampling to 10x10m

**Optical**

Atmospheric correction → Radiometric calibration → Principal component analysis (PCA) → S1 and S2 resampling to 10x10m

Select sample labels from preprocessed SAR images → Extract multi-scale features in the SAR image of S1 → Feature fusion in FC layer

Select sample labels from preprocessed optical images → Extract the high-level feature information in the optical image of S2 → Feature fusion in FC layer

Feature fusion in FC layer → Sea ice classification → Accuracy assessments

Sea ice classification → Compare classification results with other algorithms

## Experiment analysis

**Figure 1.** Sea ice classification framework for the heterogeneous data fusion.

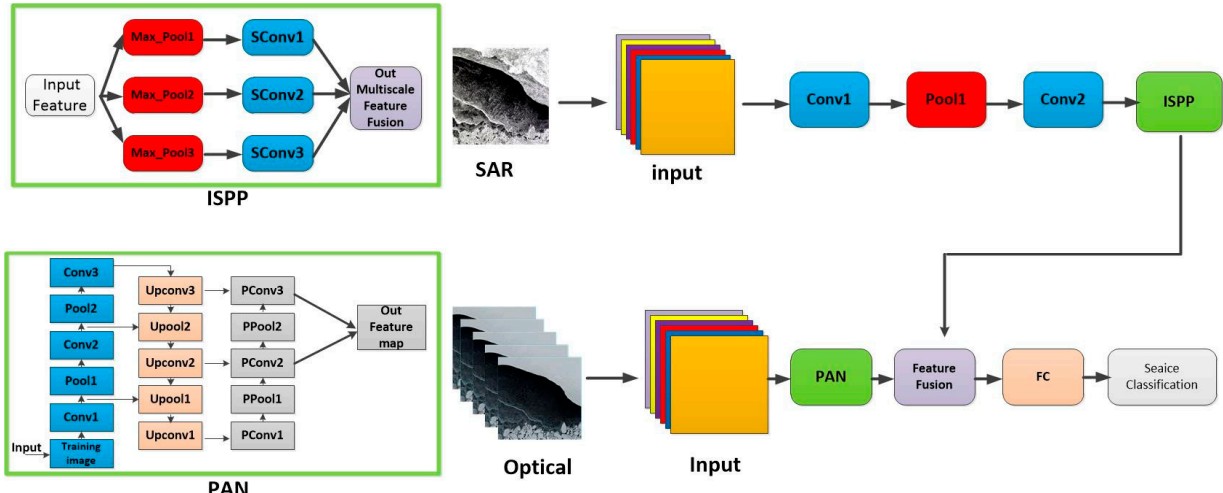

**Figure 2.** Network model of the heterogeneous data fusion.

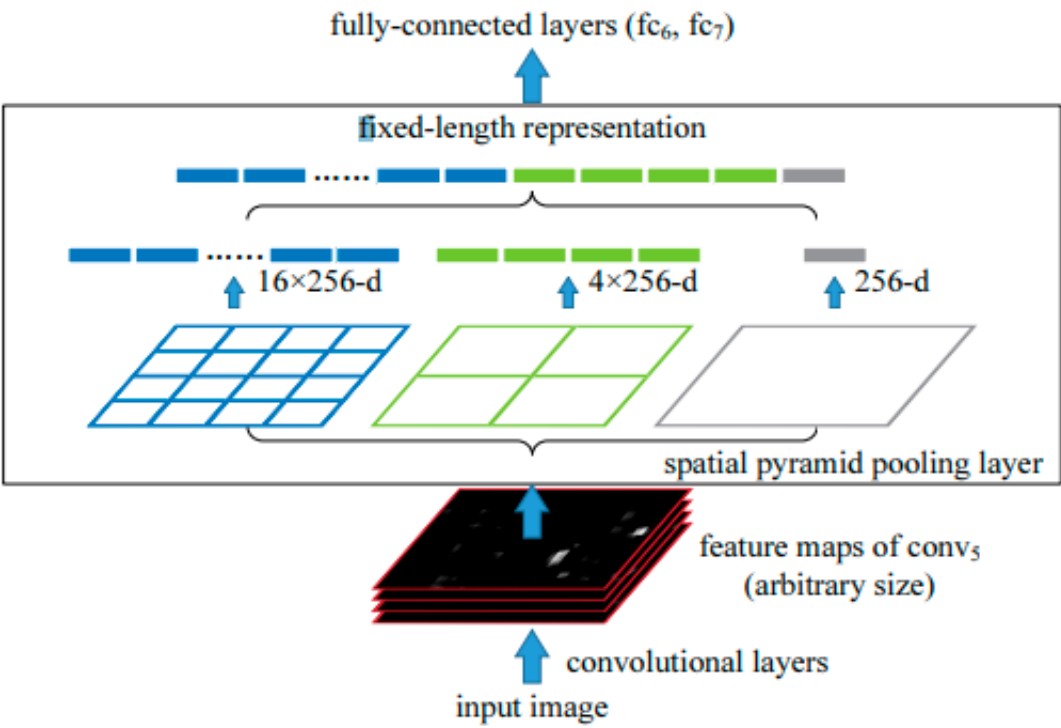

**Figure 3.** Spatial pyramid network structure.

Because SAR images have rich texture information, the SPP module can be used to extract multi-scale features. In this paper, an ISPP model is put forward using the ideas of the SPP model in addition to increasing the depth of the network. The convolution operation is carried out for three max-pooling feature maps, finally the three convoluted features are fused. As shown in Figure 4, the improved spatial pyramid pooling model can extract more high-level features, which can further improve the classification accuracy of sea ice.

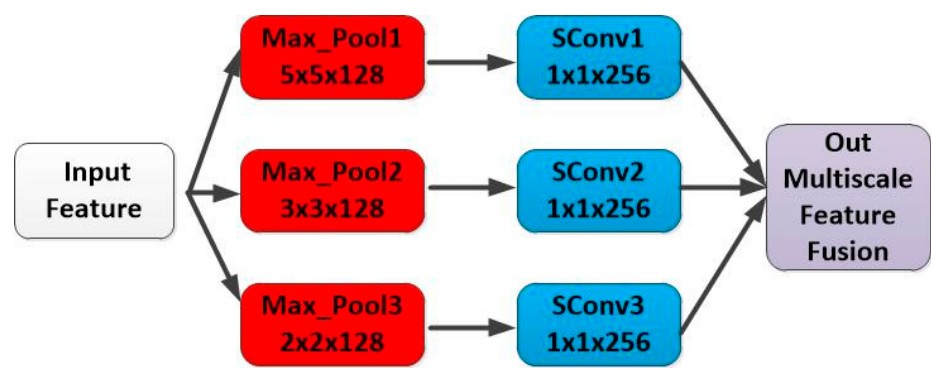

**Figure 4.** Improved spatial pyramid pooling model.

2.2.2. PANet Network Model

PANet is a path aggregation network that aims to promote the flow of feature information and connect a feature grid with all feature layers so that useful information in each feature layer can be directly transmitted to the subsequent sub-network and the feature of each layer can be fully utilized. As shown in Figure 5 [12], PANet has three convolution network modules. Module C1 is the process of input image sampling, and Module C3 subsamples module C2 and links the characteristics of the flow at the same time, thus increasing the speed of the transfer path in Figure 5. The transfer path is the green dotted

line, which contains less than 10 layers that are spread across the convolution. In contrast, CNN in FPN has a long path (the red dotted line in Figure 5), which goes from the bottom to the top and through more than 100 layers.

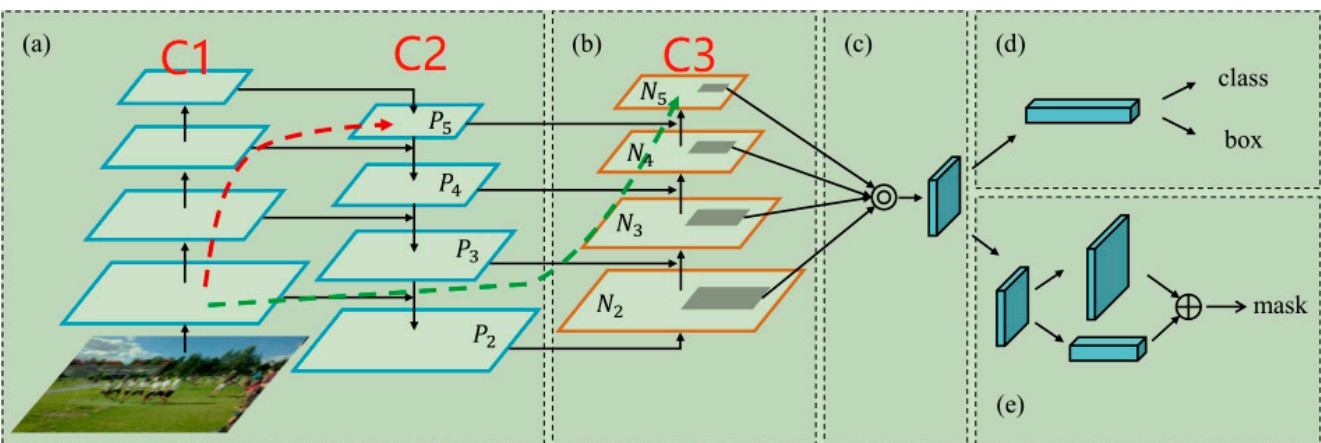

**Figure 5.** PANet partial network diagram. (**a**) FPN backbone. (**b**) Bottom-up path augmentation. (**c**) Adaptive feature pooling. (**d**) Box branch. (**e**) Fully-connected fusion.

In addition to obtaining deep-level features, the convolutional neural network can also extract low-level and middle-level features. Features extracted from each convolutional layer express different information [21]. The low-level layer lays emphasis on the contour, color, and other information, whereas the high-level layer lays emphasis on abstract features. By analyzing the features of each layer of the optical image, it was found that the features of the middle and high layers have a great influence on the classification results. This paper proposes an optical image feature extraction method based on the idea of a PANet network that can extract multi-level features and be fully utilized by the PANet network. The specific model is shown in Figure 6 below.

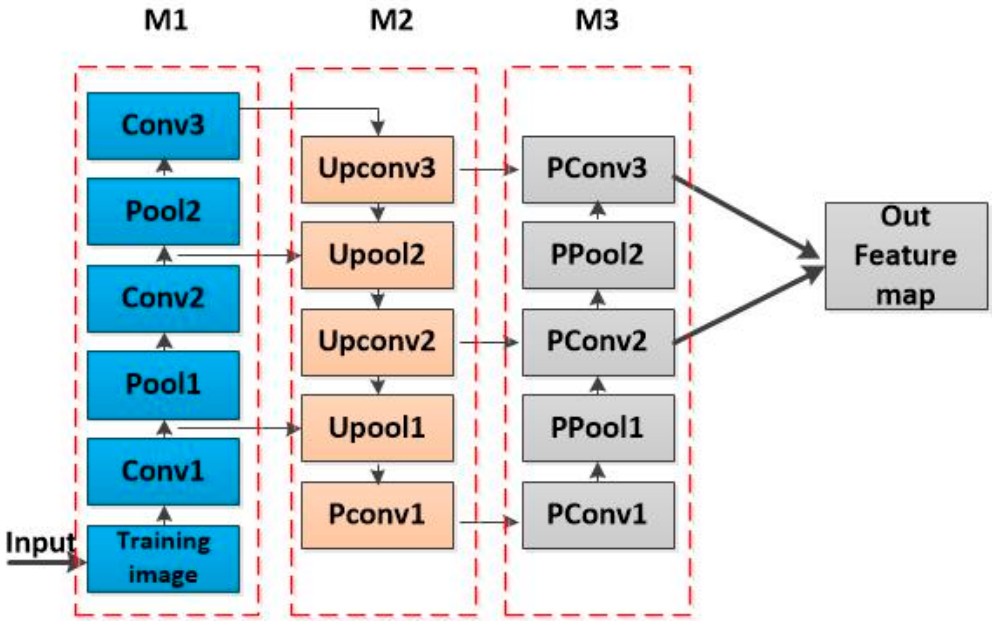

**Figure 6.** The PANet network used in the method presented in this paper.

*2.3. Algorithm Implementation Process*

After the description of the above framework, the specific implementation process of its algorithm can be shown as follows (Algorithm 1).

---

**Algorithm 1**. The algorithm process in this paper.

---

**Start**
**Input**: raw SAR data, optical data
**A. SAR image feature extraction**
(1) SAR images are preprocessed by thermal sound removal, speckle removal, geometric correction, resampling, and normalization;
(2) In the SAR image of step (1), the pixel point corresponding to each label is taken as the center and the image block with a space field size of **27 × 27** is selected as the input sample of this point;
(3) SAR image samples are only used as training samples. The fused SAR image training samples are selected according to a certain proportion of the optical training samples, and then the SAR training samples are inputted into the ISPP network;
(4) Multi-scale feature F1 of the SAR image in ISPP network is obtained; and
(5) Feature extraction of the SAR image is completed.
**B. Optical image feature extraction**
(6) Atmospheric correction, radiometric calibration, and normalization are performed on the optical images;
(7) Principal component analysis (PCA) is performed on the basis of (6) an image to extract the first principal component;
(8) Step (2) is repeated to select the input sample of the optical image;
(9) The input images obtained from (8) are divided into training samples and the samples are tested, with a ratio of 2:8;
(10) The training sample is inputted into the PANet network;
(11) The middle- and high-level characteristic information F2 of the optical image is obtained in the PANet network; and
(12) Feature extraction of the optical image is completed.
**C. Feature fusion of heterogeneous data**
(13) The characteristic sizes of one-dimensional F1 features and F2 features are made to be consistent;
(14) The features of (13) are fused;
(15) After fusion, the features are inputted into the full connection layer; and
(16) The Softmax classifier is introduced for classification.
**Output**: confusion matrix, overall accuracy, Kappa coefficient

---

**End**

---

## 3. Experimental Results and Discussion

In order to verify the effectiveness of the experimental method presented in this paper, two sets of sea ice image data at different times were used for evaluation and compared to single-source data network models, such as SVM, 2D-CNN, 3D-CNN, and PANet, as well as with classification methods of fusion models, such as the two-Branch CNN [20] and deep fusion [21]. The experimental results were evaluated in terms of the overall accuracy (OA) and Kappa values.

*3.1. Research Area and Data Preprocessing*

Hudson Bay, located in Northeastern Canada, is one of five hot spots for sea ice monitoring in the Canadian Ice Center (CIS). Sentinel-1 (S1) and Sentinel-2 (S2) are Earth Observation satellites FROM the European Space Agency Copernicus Project. S1 carries a C-band synthetic aperture radar, and S2 is a high-resolution multi-spectral imaging satellite carrying a multi-spectral imager (MSI).

The experimental data were downloaded from the European Space Agency (ESA) official website, wherein the SAR dataset of S1 is the Ground Range Detected (GRD) product, and the optical dataset of S2 is the Level-1C (L1C) product. Two datasets from partial areas of Hudson Bay were selected for analysis. Each dataset contained S1 and S2

images, which were acquired for the same area at the same time. Among them, the first dataset (Data 1) was from 6 February 2020, and the second dataset (Date 2) was from 6 April 2020. The geographical location of the study area is shown in Figure 7.

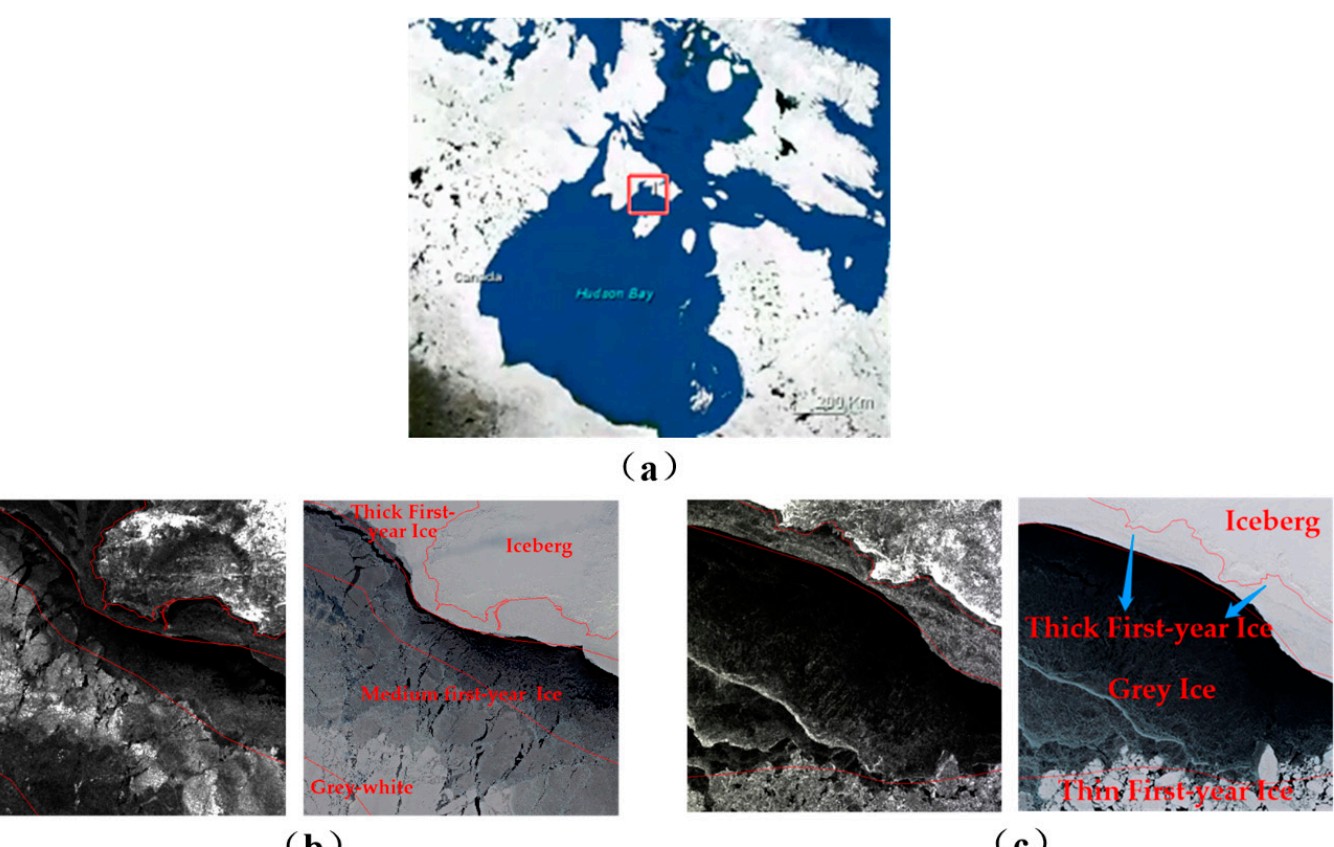

**Figure 7.** (**a**) Hudson Bay area. (**b**) The left and right images are SAR and optical RGB images, respectively, from 6 February, 2020. (**c**) The left and right images are SAR and optical RGB images, respectively, from 6 April 2020.

Before the experiment, the selected remote sensing images were preprocessed. In the S1 images, spot filtering, radiometric calibration, and geographic correction were performed. The S2 images were corrected by atmosphere and radiation. Due to the different resolutions of the S1 and S2 images, it was necessary to resample the resolution of the S1 images to 10 m, and resample the bands with 20 m and 60 m resolutions in the S2 images to 10 m. Since optical images contain multiple bands, in order to reduce the calculation cost, the remote sensing classification model adopts the two-dimensional convolutional neural network. At the same time, in order to obtain as much information on the optical image as possible, principal component analysis (PCA) is used to reduce the dimension of the optical image. After dimension reduction, the image retains the main spectral features and also contains the spatial information. SAR images and optical images are normalized, and the Min-Max normalization method is adopted. The normalization formula is as follows:

$$\text{Result} = (\text{DN} - \text{DNmin})/(\text{DNmax} - \text{DNmin}) \tag{1}$$

In the formula, Result is the normalized result value, DN is the pixel value of the original image, and DNmin and DNmax are the minimum and maximum value of the pixel in all bands, respectively.

In remote sensing imaging after preprocessing, according to the Canadian ice conditions provided by an ice chart, the first dataset types were divided into medium first-year

ice, gray-white ice, thin first-year ice, and icebergs. The second dataset was divided into thick first-year ice, gray ice, thin first-year ice, and icebergs through manual annotation tag sample production from the label sample library. The ice chart link is as follows: https://iceweb1.cis.ec.gc.ca/Archive/page1.xhtml (accessed on 5 January 2021).

The number of samples of each type of label selected from the two sets of optical image data according to ice type is shown in Table 1.

**Table 1.** Total number of labels in the S2 data sample.

| Number | Color | Class | Data1 | Data2 |
|---|---|---|---|---|
| | | | S2 | S2 |
| 1 | | Medium first-year | 4257 | - |
| 2 | | Gray-white | 5033 | - |
| 3 | | Thin first-year | - | 4014 |
| 4 | | Gray | - | 4074 |
| 5 | | Thick first-year | 4091 | 3139 |
| 6 | | Iceberg | 4629 | 3078 |

The model training sampling was conducted in accordance with the types of sea ice label samples, and the concrete steps are as follows: For each pixel within a certain range, it is highly likely that the space within the neighborhood of the adjacent pixels belongs to the same category, so it is centered in the M × M neighborhood and all pixels in the neighborhood are taken as input data. The final formation of a block of data with the size M × M × B, as model training samples, spectral information, and spatial information, can be used simultaneously. As shown in Figure 8, a square represents a pixel, take the 3 × 3 image size as an example, taking the pixel m as the center, its spatial neighborhood is m1~m8, and the pixel m and its spatial neighborhood belong to the same category in a great probability. So the image block [$m_1$, $m_2$, $m_3$, $m_4$, $m_5$, $m_6$, $m_7$, $m_8$] is taken as the training sample of pixel m. In this way, we could obtain an image size of n × n as the model sample sample. In the experiment presented in this paper, we used an image block of 23 × 23 for training.

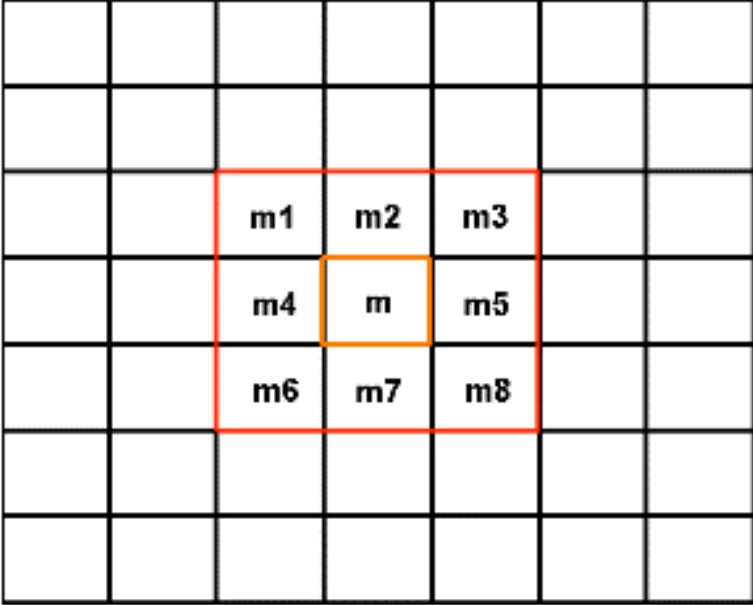

**Figure 8.** Training sample extraction method.

*3.2. Experimental Setup*

In the experiment, we used multi-spectral optical remote sensing sea ice data (S2) to carry out an experimental analysis, and the ratio of the training samples to test samples was 2:8. Meanwhile, the feature information from the SAR data (S1) was fused in the experiment to further improve the sea ice classification accuracy. In the proposed method, the multi-scale features information of SAR images was extracted with the ISPP network, and the PANet network was used to extract the mid-level and high-level features of optical image information. Then these features from heterogeneous data were fused and inputted into the Softmax classifier. The test samples were classified by the trained classifier, and finally the overall classification accuracy was calculated by using a confusion matrix. The overall classification accuracy of sea ice in the experiment was the average of the classification results of five experiments.

### 3.2.1. ISPP Model Structure

The ISPP network was used to extract the features of the SAR image of sea ice in the experiment. The specific network structure and parameters are shown in Table 2 below. The model consists of two layers of convolution and one layer of pooling. The training sample size was $27 \times 27$, the number of convolution kernels at the first layer was 64, the stride size of the convolution operation was $1 \times 1$, the number of convolution kernels at the second layer was 128, and the stride size of the convolution operation was $1 \times 1$. After two convolutions, the feature map was inputted into the ISPP module for feature extraction at three different scales. The stride sizes of the three pooling layers were $2 \times 2$, $4 \times 4$, and $8 \times 8$, respectively. Then the feature map after each pooling was further convoluted to extract deep semantic information, and the obtained deep information was featured by feature fusion. During the whole training process, the learning rate of the model was 0.001, the dropout value was 0.5, and the activation function used was Rectified Linear Unit (ReLU).

**Table 2.** Network parameters for extracting SAR image features.

| | Input | Conv1 | Pool1 | Conv2 | Max-Pool (ISPP) | | | SConv (ISPP) | | |
| --- | --- | --- | --- | --- | --- | --- | --- | --- | --- | --- |
| | | | | | MaxPool1 | MaxPool2 | MaxPool3 | SConv1 | SConv2 | SConv3 |
| **Kernel Size Strides** | - | 64 | - | - | - | - | - | 256 | 256 | 256 |
| **Map size** | - | [1,1] | [2,2] | [2,2] | [2,2] | [4,4] | [8,8] | [1,1] | [1,1] | [1,1] |
| **Kernel Size Strides** | $27 \times 27$ | $25 \times 25$ | $12 \times 12$ | $10 \times 10$ | $5 \times 5$ | $3 \times 3$ | $2 \times 2$ | $1 \times 1$ | $1 \times 1$ | $1 \times 1$ |

### 3.2.2. PANet Model Structure

Multi-layer feature extraction and a fusion model using the PANet network were used in the experiment for the optical image of sea ice. The fusion model consists of three modules, two subsampling modules (modules M1 and M3) and one upsampling module (module M2). The features of each layer are connected, and the middle and high features in the network are finally fused. The network structure and parameters are shown in Table 3. The size of the training image input in the experiment was still $27 \times 27$. In module M1, feature extraction was carried out on the input training image. The module includes three convolutional layers and two pooling layers. The stride of the three convolution layers was $1 \times 1$; the number of convolution kernels was 32, 64, and 128 for M1, M2, and M3, respectively; and the stride of the two pooling layers was $2 \times 2$. In module M2, upsampling is mainly carried out by module M1 and connects the features extracted from module M1. Upsampling methods included deconvolution and unpooling, with two layers of deconvolution and two layers of unpooling, and the stride was $1 \times 1$ and $2 \times 2$, respectively. In module M3, the features of PConv1 were obtained by subsampling, and the feature information extracted by module M2 was connected. The module contains two pooling layers and two convolution layers.

**Table 3.** PANet network parameters.

|  | Layers | Feature Map Size | Strides | Number of Kernels |
|---|---|---|---|---|
| **M1** | Conv1 | $26 \times 26$ | [1,1] | 32 |
|  | Pool1 | $13 \times 13$ | [2,2] | - |
|  | Conv2 | $12 \times 12$ | [1,1] | 64 |
|  | Pool2 | $6 \times 6$ | [2,2] | - |
|  | Conv3 | $5 \times 5$ | [1,1] | 128 |
| **M2** | PConv1 | $26 \times 26$ | - | 32 |
|  | Upool1 | $26 \times 26$ | [2,2] | - |
|  | Upconv2 | $13 \times 13$ | [1,1] | 32 |
|  | Upool2 | $12 \times 12$ | [2,2] | - |
|  | Upconv3 | $6 \times 6$ | [1,1] | 64 |
| **M3** | PConv1 | $26 \times 26$ | [1,1] | 32 |
|  | PPool1 | $13 \times 13$ | [2,2] | - |
|  | PConv2 | $12 \times 12$ | [1,1] | 64 |
|  | PPool2 | $6 \times 6$ | [2,2] | - |
|  | PConv3 | $5 \times 5$ | [1,1] | 128 |

*3.3. Analysis of the Experimental Parameters*

3.3.1. Influence of the PANet Model Training Sample Size

The training sample size is an important factor affecting the classification accuracy of the model. The selection of training sample size comprehensively considers the spatial information contained in the sample and the depth of the network model. The larger the size of the training sample is, the more spatial information it contains, which can improve the depth of convolution network and mine more feature information. However, because the surrounding samples may not belong to this category, it will also bring some errors. The smaller the size of training sample, the smaller the error caused by adjacent pixels, but the smaller training sample size contains less spatial information. At the same time, due to the size limitation of training sample, it will reduce the number of layers of convolution network, and it is difficult to obtain more deep information, on the contrary, it will reduce the classification accuracy. Considering the above factors, five training sample sizes $19 \times 19$, $21 \times 21$, $23 \times 23$, $25 \times 25$, $27 \times 27$ for the sea ice classification were evaluated. The experimental results show that the training sample size of $27 \times 27$ can obtain better classification results, so this training sample size is chosen in this paper, as shown in Table 4.

**Table 4.** Influence of different training sample sizes on classification accuracy.

|  | Data 1 | | Data 2 | |
|---|---|---|---|---|
|  | **OA (%)** | **Kappa$\times$100** | **OA (%)** | **Kappa$\times$100** |
| **19$\times$19** | 89.89 | 88.71 | 90.09 | 88.97 |
| **21$\times$21** | 91.13 | 89.84 | 91.15 | 89.65 |
| **23$\times$23** | 92.74 | 90.67 | 91.85 | 89.93 |
| **25$\times$25** | 93.32 | 91.01 | 92.33 | 90.27 |
| **27$\times$27** | 93.76 | 91.94 | 93.07 | 91.32 |
| **29$\times$29** | 93.45 | 91.53 | 92.87 | 91.07 |

3.3.2. Influence of the Convolution Kernel Size of the PANet Model

The convolution operation is the main way to extract the features of the CNN model, and the size of the convolution kernel plays an important role in the performance evaluation of the model. As shown in Figure 2 above, in the PANet network, the model conducts multiple upsampling and subsampling processes, connects the features of different layers, and finally extracts the features of the middle- and high-level images.

Based on the above network model, experiments were conducted on S2 data in Data 1 and Data 2. In the experimental comparison of the size of the input model, the final size

selected in the experiment was 27 × 27. In view of the model structure and the size of the input training image, experiments were carried out on 2 × 2 and 4 × 4 convolution kernel sizes.

During the experiment, the training samples were randomly selected. In order to avoid the contingency in the final experimental results, five experiments were conducted for each dataset, and the average value was taken as the overall classification result. Table 5 shows the classification accuracy results obtained when different convolution kernel sizes were adopted in the two datasets.

**Table 5.** Influence of different convolution kernel sizes on classification accuracy.

|  | Data 1 | | Data 2 | |
| --- | --- | --- | --- | --- |
|  | **OA (%)** | **Kappa×100** | **OA (%)** | **Kappa×100** |
| **2×2** | 93.76 | 91.94 | 93.07 | 91.32 |
| **4×4** | 93.12 | 91.21 | 92.64 | 89.99 |

It can be seen from Table 5 that the classification accuracy of the model varied with the convolution kernel size. In the two sets of experimental data, when the convolution kernel size was 2 × 2, the model achieved good classification accuracy and the overall accuracy of Data 1 and Data 2 reached 93.76% and 93.07%, respectively. In the following experiments, the convolution kernel size was 2 × 2.

### 3.3.3. Influence of the Number of Samples for SAR Data Fusion

Due to the rich texture information contained in SAR data, SAR image feature information can be used to effectively improve the classification performance. In the experiment, SAR data and optical data were trained separately to extract features. First, the ISPP model was used to extract multi-scale features from SAR image data, and the extracted features were processed one-dimensionally. In addition, the PANet model was used to extract the feature of the middle level and high level of the optical images, and these features were also one-dimensionally processed and fused with the multi-scale features of SAR images. Finally, they were inputted into the full connection layer to classify the optical images.

In the feature fusion experiment, training samples of SAR images were randomly selected from the S1 sample base for training. An optical image of the training sample was randomly selected from the S2 sample for training, and the test sample was the optical images in the dataset. In the following experiments, SAR training label features and optical label features were extracted and fused. Two different kinds of characteristics from the source data integration analysis concerning the result of the sea ice classification are shown in Table 6 below.

**Table 6.** Proportion of SAR training samples (S1) and optical image training samples (S2).

| Data Set | Data | 1 | 2 | 3 | 4 | 5 | 6 | 7 | 8 | 9 | 10 |
| --- | --- | --- | --- | --- | --- | --- | --- | --- | --- | --- | --- |
| Data 1 | S1 | 3585 | 1792 | 1195 | 896 | 717 | 597 | 512 | 448 | 398 | 358 |
|  | S2 | 3585 | 3585 | 3585 | 3585 | 3585 | 3585 | 3585 | 3585 | 3585 | 3585 |
| Data 2 | S1 | 2858 | 1426 | 852 | 714 | 571 | 476 | 408 | 357 | 317 | 258 |
|  | S2 | 2858 | 2858 | 2858 | 2858 | 2858 | 2858 | 2858 | 2858 | 2858 | 2858 |
| Ratio | S1:S2 | 1:1 | 1:2 | 1:3 | 1:4 | 1:5 | 1:6 | 1:7 | 1:8 | 1:9 | 1:10 |

The classification accuracy and Kappa coefficient results, obtained with different fusion ratios listed in Table 6, are shown in Figure 9 below, and the classification accuracy is the average of the results of five experiments.

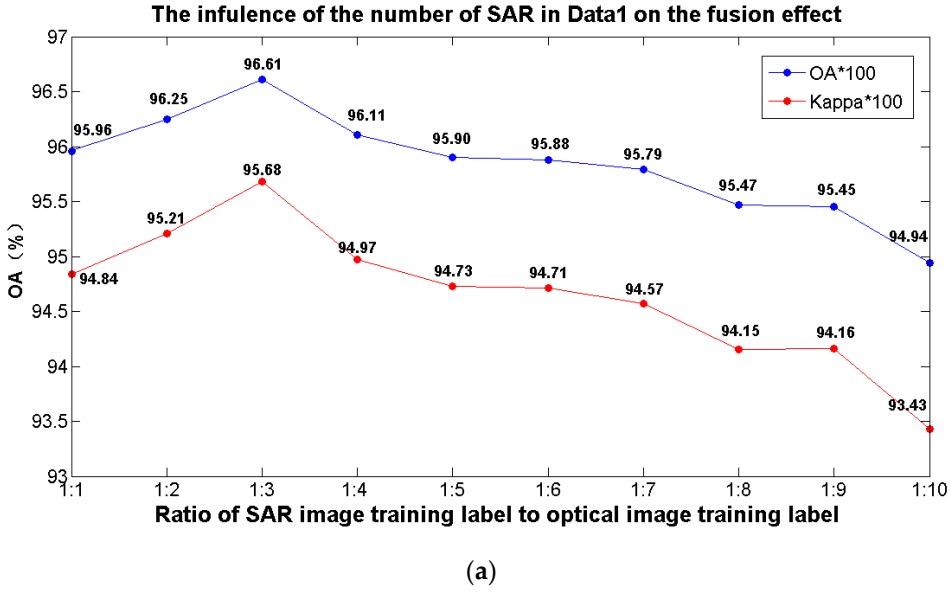

(**a**)

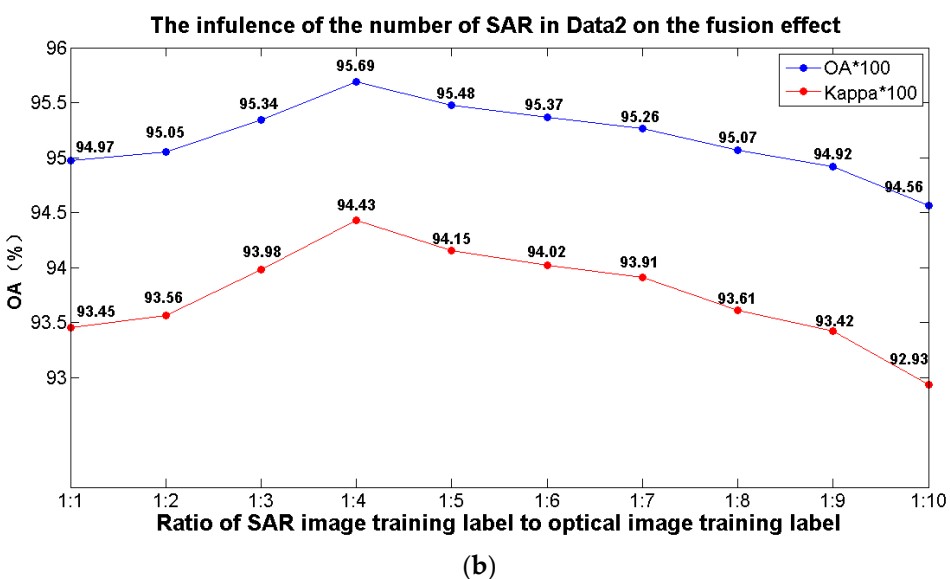

(**b**)

**Figure 9.** Classification performance of different training sample proportions of heterogeneous data.

In the above experiment, multi-spectral optical remote sensing sea ice data (S2) were used to carry out the experimental analysis, and the ratio of the training samples to the test samples was 2:8. Meanwhile, the characteristic information from the SAR data (S1) was fused in the experiment to further improve the sea ice classification accuracy. Figure 9 shows the results of the comparative analysis when different proportions of SAR training samples were fused in the experiment. As can be seen from Figure 9, when the number of training samples of the optical image remained unchanged, the proportion of the training samples of the fused SAR image was adjusted. After the feature fusion of the two kinds of heterogeneous data, the final classification accuracy of sea ice was different. In Data 1, when the ratio of SAR training samples to optical training samples was 1:3, the overall accuracy reached the highest value of 96.61%, and the Kappa coefficient was 95.68, which was 2.85 percentage points higher than the accuracy of 93.76% when using optical data alone for classification. In Data 2, when the ratio of the number of SAR training samples to the number of optical image training samples was 1:4, the classification accuracy of sea ice was the best, and the overall accuracy reached 95.69%, which was 2.63 percentage points higher than the classification accuracy of 93.07% using optical data alone.

The abovementioned experimental results show that compared to using optical images alone to classify sea ice, the classification accuracy was significantly improved after fusing the feature information of the SAR image. In addition, by fusing different proportions of SAR image samples, the improvement of classification accuracy was different. Too many training samples of a fused SAR image reduce the generalization ability of the model, and too few samples of a fused SAR image cannot achieve the desired effect. By choosing the appropriate proportion of fusion samples, we can obtain better sea ice classification accuracy.

### 3.4. Analysis of Experimental Results

#### 3.4.1. Comparison with Other Image Classification Methods

Table 7 shows the comparative analysis results of the method presented in this paper and other typical image classification fusion methods using single-source data, among which several commonly used classification methods only use optical data for classification. It can be seen from the experimental results that the method presented in this paper achieved the best classification results compared to other methods, and the overall accuracy was 96.61% and 95.69% for the two datasets, respectively. It was 6.56% and 5.95% higher, respectively, than that of the SVM. This is because the SVM model mainly extracts the shallow features, which limits the improvement of its classification accuracy. The 2D-CNN model mainly classifies using high-level feature information but does not make full use of middle-level feature information and spatial information. The accuracy was 91.78% and 91.06%, respectively, which was 4.83% and 4.53% lower, respectively, than the method presented in this paper. In the 3D-CNN model, spatial and spectral information can be extracted simultaneously, which can effectively improve the classification accuracy. The accuracy for the two datasets was 93.65% and 93.15%, respectively. The PANet network utilizes middle-level and high-level feature information, but, like 2D-CNN, it does not extract much spatial information. The overall classification accuracy was 93.76% and 93.07% for the two datasets, respectively. Compared to the commonly used classification methods, the method proposed in this paper showed the best classification effect in experiments due to the multi-scale feature of SAR images and the middle–high-level feature of optical images.

**Table 7.** Comparison of the methods presented in this paper and the classification methods based on single-source data.

| | | Medium First-Year Ice | Gray-White Ice | Thin First-Year Ice | Gray Ice | Thick First-Year Ice | Iceberg | OA | Kappa×100 |
|---|---|---|---|---|---|---|---|---|---|
| Data 1 | SVM | 92.31 | 89.87 | - | - | 90.23 | 89.43 | 90.05 | 89.12 |
| | 2D-CNN | 92.53 | 91.34 | - | - | 91.91 | 90.97 | 91.78 | 90.81 |
| | 3D-CNN | 94.76 | 92.98 | - | - | 93.83 | 92.47 | 93.65 | 92.09 |
| | PANet | 94.55 | 93.02 | - | - | 94.30 | 92.71 | 93.76 | 91.94 |
| | Proposed | 96.86 | 96.26 | - | - | 96.71 | 97.02 | **96.61** | **95.68** |
| Data 2 | SVM | - | - | 90.21 | 90.45 | 89.67 | 89.12 | 89.74 | 88.26 |
| | 2D-CNN | - | - | 92.12 | 91.23 | 90.02 | 90.34 | 91.06 | 90.24 |
| | 3D-CNN | - | - | 93.89 | 94.01 | 92.12 | 92.34 | 93.15 | 91.34 |
| | PANet | - | - | 94.21 | 93.78 | 92.78 | 92.31 | 93.07 | 91.32 |
| | Proposed | - | - | 96.11 | 95.94 | 95.37 | 95.55 | **95.69** | **94.43** |

#### 3.4.2. Comparison of Different Fusion Methods

In order to further verify the performance of the proposed method in multi-source remote sensing data fusion classification, the proposed method was compared to other fusion methods. The experimental results are shown in Table 8. The SVM (S1+S2) method was used to train an SAR image and optical image after the training samples were mixed. The two-branch CNN utilizes the tow convolutional neural network to extract the characteristics of two kinds of heterogeneous data. The deep fusion model uses multiple networks to extract features from heterogeneous data.

**Table 8.** Comparison of different fusion methods.

| Class | Data1 | | | | Data2 | | | |
|---|---|---|---|---|---|---|---|---|
| | SVM (S1 + S2) | Tow-Branch CNN | Deep Fusion | Prop-osed | SVM (S1 + S2) | Tow-Branch CNN | Deep Fusion | Prop-osed |
| Medium first-year ice | 93.04 | 96.89 | 96.43 | 96.86 | - | - | - | - |
| Gray-white ice | 90.14 | 96.03 | 96.32 | 96.26 | - | - | - | - |
| Thin first-year ice | - | - | - | - | 91.67 | 95.55 | 95.92 | 96.11 |
| Gray ice | - | - | - | - | 91.23 | 95.32 | 95.43 | 95.94 |
| Thick first-year ice | 92.31 | 95.96 | 95.78 | 96.71 | 90.12 | 94.89 | 95.71 | 95.37 |
| Iceberg | 92.78 | 96.84 | 96.59 | 97.02 | 90.88 | 94.82 | 94.51 | 95.55 |
| OA | 92.50 | 96.28 | 96.31 | 96.61 | 90.96 | 95.28 | 95.42 | 95.69 |
| Kappa×100 | 90.03 | 95.12 | 95.24 | 95.68 | 89.87 | 94.14 | 94.37 | 94.43 |

As can be seen from the experimental results, compared to other methods, the proposed method achieved the best classification results. The overall classification accuracy for the two datasets was 96.61% and 95.69%, respectively, and the Kappa coefficient was 95.68 and 94.43, respectively. Compared to the SVM method, the accuracy of the SVM method was improved by 4.11% and 4.63%, respectively. Because the SVM extracted features were shallow features, it was difficult to obtain a higher classification accuracy. The two-branch CNN mainly extracted high-level semantic features, which limited its classification accuracy to 96.28% and 95.28%, respectively. The accuracy of the deep fusion model was 96.31% and 95.42%, respectively, due to the lack of low-level features. The method proposed in this paper, on the one hand, fuses the different features from different data sources; on the other hand, it fuses the multi-scale and multi-level features for different data sources to further improve the classification effect and obtain the highest classification accuracy.

Compared to different fusion methods, the fusion algorithm of multi-source features proposed in this paper achieved good results, the classification accuracy of the two sets of data was 96.61% and 95.69%, respectively. In order to verify the validity of the proposed algorithm and better show the sea ice classification effect of this method, the results of the above heterogeneous fusion model were visualized as shown in Figure 10. It can be seen from the figure that the classification results of the proposed method were in good agreement with the original image.

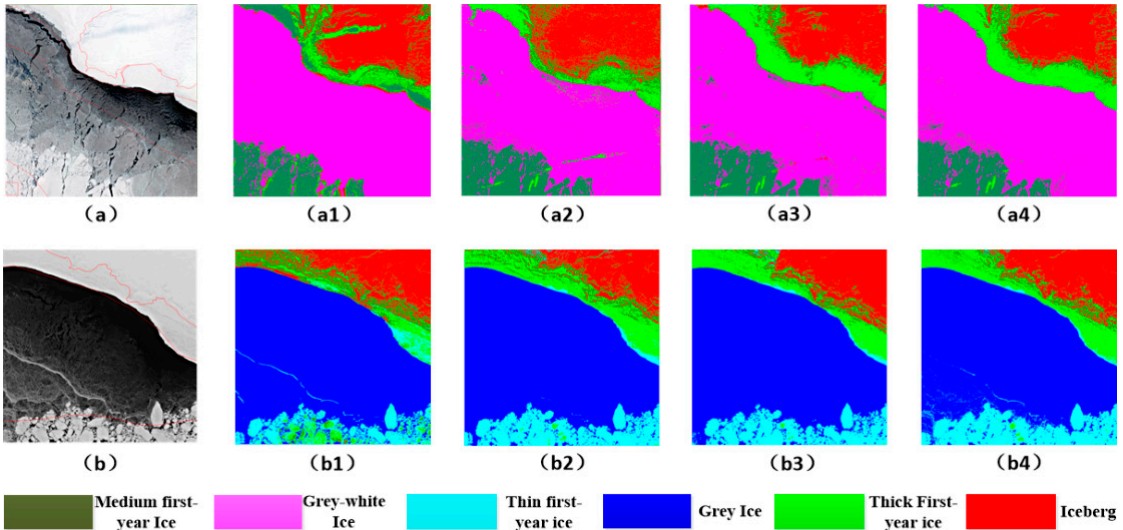

**Figure 10.** (**a**,**a1**–**a4**) are a false color image of Data 1, an SVM visualization image, a two-branch CNN visualization image, a deep fusion visualization image, and a visualization image of the method in this paper, respectively. (**b**,**b1**–**b4**) are a false color image of Data 2, an SVM visualization image, a two-branch CNN visualization image, a deep fusion visualization image, and a visualization image of the method in this paper, respectively.

In terms of time efficiency, the results of the comparison of the proposed method with other fusion methods are shown in Table 9. All experiments were run on the same equipment, and the average running time of the five experiments was taken as the result. Since SVM (S1 + S2) is a shallow learning method, it had the best performance in terms of time efficiency compared to the deep learning methods. The deep fusion network uses three deep frameworks (two CNNs and one DNN), and it took a relatively long time to train the model.

**Table 9.** Training time of the different methods (unit: seconds).

|  | SVM (S1 + S2) | Tow-Branch CNN | Deep Fusion | Proposed |
|---|---|---|---|---|
| **Data 1** | 102.73 | 501.25 | 582.37 | 509.95 |
| **Data 2** | 87.62 | 391.33 | 491.16 | 393.58 |

Both the two-branch CNN and the proposed method in this paper adopt two deep network frameworks, which had little difference in terms of time efficiency. However, the proposed method achieved better classification accuracy.

## 4. Conclusions

In this article, SAR data and the optical characteristics of a remote sensing data fusion are applied in the classification of sea ice, making full use of the abundant sea ice texture features in SAR data and optical remote sensing images to provide high-resolution spectral characteristics, design a sea ice deep learning model to extract heterogeneous multi-scale feature and multi-level feature information, and improve classification accuracy. Through the analysis and comparison to other classical image classification methods and heterogeneous data fusion methods, this paper proposes a method to obtain a better sea ice classification result, which provides a new method and idea for remote sensing sea ice image classification using heterogeneous data fusion. The specific contributions are as follows:

(1)　Optical remote sensing data are rich in spectral features, and a SAR sensor can obtain abundant ground texture information. Heterogeneous data fusion can overcome the limitations of single-source data and make full use of the characteristic information of data from different data sources in order to realize complementary advantages, providing a new way of thinking of the classification of remote sensing sea ice images.

(2)　Based on the advantages of convolution neural networks in extracting deep features, a deep learning and heterogeneous data fusion method for sea ice image classification designed for the convolution neural network structure of SAR images and optical images, the extraction of heterogenous multi-scale features and multi-level features, and the implementation of sea ice image classification using feature level fusion, the sea ice image classification accuracy is obviously increased.

(3)　The training sample size of the deep learning model, the size of the convolution kernels, and the heterogeneous data integration of different data fusion ratios impact the sea ice classification accuracy. To further improve the learning effect of deep learning models and thus the sea ice classification accuracy, the parameters of the deep learning model were analyzed and compared in terms of the size of the training sample, the size of the convolution kernel and the fusion ratio of SAR data, so as to further improve the accuracy of sea ice classification.

In addition, because the SAR sensor can penetrate through clouds and mist, it is not affected by clouds or mist, whereas optical remote sensing is be affected by the interference of clouds and mist. Through heterogeneous data fusion, data complementary can be realized, and the advantages of heterogeneous data can be fully utilized to further expand the scope of sea ice detection and improve the accuracy of sea ice detection, which is our next research content.

**Author Contributions:** Y.H. and Z.H. conceived and designed the framework of the study. Y.L. completed the data collection and processing, and completed the experiment. Y.H. and S.Y. completed the algorithm design and the data analysis. Y.Z. was the lead author of the manuscript with contributions by Y.L., Y.H., and J.W. All authors have read and agreed to the published version of the manuscript.

**Funding:** This work was supported by the National Natural Science Foundation of China (Grant Nos.41871325, 61806123), and the Open Project Program of the Key Laboratory of Fisheries Information of the Ministry of Agriculture.

**Data Availability Statement:** Enquiries regarding experimental data should be made by contacting the first author.

**Acknowledgments:** The authors would like to thank the European Space Agency (ESA, data https://sentinel.esa.int/web/sentinel/home (accessed on 5 January 2021)) for providing satellite images on Polar.

**Conflicts of Interest:** The authors declare no conflict of interest.

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
