# Peer review of "Sea Ice Image Classification Based on Heterogeneous Data Fusion and Deep Learning"

_remotesensing, doi:10.3390/rs13040592_

Round 1

Reviewer 1 Report

In this paper, a method of sea ice image classification based on deep learning and heterogeneous data fusion is proposed. By utilizing the advantages of convolutional neural network in-depth feature extraction, a deep learning network structure is designed for SAR images and optical images, and sea ice image classification is realized through feature extraction and feature-level fusion of heterogeneous data. The paper is interesting and well-written but has some drawbacks:

  1. Subpart 2.2.1 is not needed. It's well-known.
  2. It'll be better to present the algorithm in subpart 2.3 graphically as a scheme.
  3. The paper should be proofread. There are a lot of typos in the text.
  4. How do the authors calculate a classification accuracy in subpart 3.3?  It's not clear.
  5. Figure 9 should be better commented on.
  6. The authors should present the performance of the proposed solution in terms of estimated time and computational complexity.

Reviewer 2 Report

The authors proposed a method to classify sea ice by simultaneously employing satellite-based SAR data and optical data. The idea is new and interesting. The observation frequency has been largely improved both for the satellite SAR sensor and optical sensor nowadays, therefore, SAR data and optical data are able to be obtained in the same day for monitoring regions of interest. Data acquisition may not be a problem for the proposed method. However, the manuscript is relatively rough. Description of the methodology and result is not adequate. Reviewer lists the comments as follows:   

  • please use abbreviations consistently in the paper.
  • The references list should follow the formatting guide of the journal. Please check again.
  • This paper has numerous grammar and language issues, which need to be addressed.
  • L56, please delete “features of”.
  • L85, please clarify the meaning of “advantages in the space”.
  • L92, please clarify the meaning of “different scale in depth”.
  • L97, the words “multi-scale” and “multi-level” frequently occur in the paper. Please clarify the meaning of them.
  • Page 7, L3, please specify all the pre-process methods rather than using “etc.” for the reader to reproduce the proposed method.
  • Page 7, is the training sample used in step (3 same with that used in step (9. And why did you set the ratio of training samples to testing samples 2:8.
  • 7. Please show S2 image in RGB scale so that readers can understand the difference between SAR sensor and optical sensor for imagery of sea ice.
  • L 247, please specify the method of normalization with equations.
  • L253, please cite the website correctly.
  • 9, fusino should be fusion, and the legends should be consistent between the two sub-figures.
  • Section 3.3.3, reviewer cannot understand the number of S1 and S2 data can be different. Suppose you have one specific location (assuming it is one pixel) in the study area with known label, the corresponding pixel in the S1 data and S2 data should be simultaneously used to train or validate the model. how can you set different ration of S1 and S2 in the model.
  • The authors discussed the effect of training sample size. However, 23*23 or 27*27 is pretty large since the pixel size if 10 m. If the size of sea ice is much smaller, could the model work well? Do you try training sample size with smaller size?
  • Please consider to show some experimental result to evidence your conclusion. For example, you show the classification of different type of sea ice with various color. It is meaningless because they are not used in the remaining of the paper. Please show the detailed experimental result with the same color show in Table 1.
  • The model proposed in the paper is complex. The computation cost should be illustrated to show the feasibility of future practical application of monitoring sea ice.
  • SAR images could be largely different when the observation condition changes. The authors use one part of the image to train the model and one part of the image for testing so that good result was achieved. However, reviewer really concern about whether similar result can be achieved when apply the trained model to satellite data obtained at some other time. Generally, more satellite data obtained at different time should be used to train the model thus the model can be more robust. please try to use more data to show the robustness of the model.
  • Reviewer think that SAR data are difficult to distinguish the sea ice type (thin ice or thick ice) since they are single band data. By fusion the S1 data and S2 data, the authors demonstrate that including the SAR can improve the classification effect. Please describe why that only using optical data cannot achieve the same result in the introduction section since multi-band optical data are more abundant.

Round 2

Reviewer 1 Report

The authors addressed all my concerns. The paper can be accepted in the present form.

Reviewer 2 Report

The manuscript has been greatly improved. Reviewer has no further questions.